

# Assessment of [222]Radon progeny loss in long tubing based on static filter measurements in the laboratory and in the field

Ingeborg Levin[1], Dominik Schmithüsen[1], Alex Vermeulen[2]

[1]Institut für Umweltphysik (IUP), Heidelberg University, 69120 Heidelberg, Germany
[2]Energy research Centre of the Netherlands (ECN), 1755LE Petten, The Netherlands, now at: ICOS ERIC - Carbon Portal, Lund, Sweden

*Correspondence to*: Ingeborg Levin (Ingeborg.Levin@iup.uni-heidelberg.de)

**Abstract.** Aerosol loss in air intake systems potentially hampers the application of one-filter systems for progeny-based atmospheric [222]Radon ([222]Rn) measurements. The artefacts are significant when air has to be collected via long sampling lines,

e.g. from elevated heights at tall tower observatories. Here we present results from a study, determining [222]Rn progeny loss from ambient air sampled via 8.2 mm inner diameter (ID) Decabon tubing (laboratory test) and from pre-installed 10 mm ID tubing at the Cabauw meteorological tower in the Netherlands. Progeny loss increased exponentially with length of the tubing, decreasing sampling efficiency to 66% for 8.2 mm ID rolled-up tubing of 200 m length. Theoretical estimation of the loss yielded a sampling efficiency of 64% for the same tubing, when taking into account turbulent inertial deposition of aerosol to

the walls as well as loss due to gravitational settling. At Cabauw tower, theoretical estimates of the loss in vertical tubing with 10 mm ID and 200 m lengths yielded a total efficiency of 75%, while we observed a slightly smaller sampling efficiency of 73%. [222]Rn progeny loss increased strongly at activity concentrations below 1 Bq m$^{-3}$. Based on our experiments, an empirical correction function for [222]Rn progeny measurements when sampling through long Decabon tubing was developed, allowing correction of respective measurements with an estimated uncertainty of 10-20% for activity concentrations above 1 Bq m$^{-3}$

and less than 10% for activity concentrations above 2 Bq m$^{-3}$.

## 1 Introduction

Soil-borne [222]Radon ([222]Rn) is widely used as tracer for atmospheric mixing and transport model validation (e.g. Jacob and Prather, 1990; Taguchi et al., 2011), as its sources are relatively well known (e.g. Nazaroff, 1992; Karstens et al., 2015), while its only sink is radioactive decay. Continuous atmospheric [222]Rn is normally measured via the activity of its short-lived

progeny, such as [218]Polonium ([218]Po) and [214]Polonium ([214]Po), which are attached to aerosols shortly after the generation of the progeny. In so-called two-filter systems, filtered air that only contains [222]Rn is flushed through a large delay chamber and the in situ produced [222]Rn progeny are collected on a second internal filter where their decay is monitored (e.g. Whittlestone and Zahorowski, 1998; Chambers et al., 2011). Alternative [222]Rn monitors, the so-called one-filter systems, collect [222]Rn progeny directly from ambient air on a static filter or automatically changing filters, and measure their decay rates on the filter.

Atmospheric [222]Rn activity concentration is then deduced from the filter activity of its progeny, making assumptions about the





disequilibrium between $^{222}$Rn and its progeny on the filter and in the atmosphere (e.g. Levin et al., 2002; Schmithüsen et al., 2016). Disequilibrium between $^{222}$Rn and its daughters depends on height above ground level, on meteorological conditions and the influence of processes causing aerosol loss by dry or wet deposition (see e.g. Jacobi and André (1963) and Porstendörfer (1994)). These non-negligible biases can be investigated by comparison with two-filter detectors (Xia et al.,

5   2010).

In addition, potential artifacts from air intake and sampling lines of one-filter systems need to be investigated before one-filter monitors could be implemented e.g. at tower stations, where long intake tubing is required. To minimize loss of aerosol-attached $^{222}$Rn daughters in the intake systems, one-filter monitors are normally used only at ground level stations, where short

intake lines are possible. This principally disqualifies one-filter systems to be applied at tall tower stations, unless the filter head can be installed directly at the level where the $^{222}$Rn daughter measurements shall be performed (e.g. at an accessible platform). Alternatively, corrections must be applied for loss of aerosol-attached $^{218}$Po or other progeny in the tubing. This, however, requires full quantitative understanding of all aerosol transport loss and deposition processes in the intake system, such as diffusion to the walls, sedimentation or inertial deposition (see e.g. Willeke and Baron (2005) for a comprehensive

discussion of these processes).

In the present work, we present the results from dedicated laboratory and field experiments, which had the aim to determine $^{222}$Rn progeny loss in long tubing. The study was motivated by the requirement to accompany greenhouse gases measurements with continuous $^{222}$Rn observations within the new European ICOS (Integrated Carbon Observation System) atmospheric

station network (https://www.icos-ri.eu/icos-research-infrastructure/icos-national-networks). Due to space restrictions at many of the ICOS tall tower sites, it may be difficult to host two-filter systems with their huge detector volumes (Whittlestone and Zahorowski, 1998). Alternatively, the Heidelberg static one-filter radon monitor (Levin et al., 2002) requires only little space, and routine maintenance work and data evaluation are less demanding than for the two-filter systems (Whittlestone and Zahorowski, 1998).


The question of compatibility of existing one- and two-filter systems for $^{222}$Rn monitoring as well as first estimates of disequilibrium have been addressed in a companion paper (Schmithüsen et al., 2016), where results from a European-wide comparison study are reported. In the current work we address potential aerosol loss in standard Decabon tubing, as this tubing material is prescribed to be used at the tall tower stations of the ICOS network. From our laboratory experiments, a correction

algorithm for $^{222}$Rn progeny loss in tubing up to 200 m length has been developed. Validation of these laboratory experiments was possible during a measurement campaign at the Cabauw tall tower station in the Netherlands, where atmospheric $^{222}$Rn observations are routinely performed with a two-filter monitor (from ANSTO, Whittlestone and Zahorowski, 1998).





## 2 Methods

### 2.1 Heidelberg static (one) filter $^{222}$Rn-Monitor (HRM)

In Heidelberg, a medium size city located in the upper Rhine valley in south-western Germany (49° 25'N, 8°41'E, 116 m a.s.l.), atmospheric $^{222}$Rn daughter activity concentrations have been continuously monitored since 1996 with a static filter

system. Measurements were conducted at the Institut für Umweltphysik, Heidelberg University building, located on the University campus in the outskirts of the city. Since 1999, ambient air is collected at about 30 m above local ground. These Heidelberg $^{222}$Rn data have been used as tracer for boundary layer mixing in a number of studies (e.g. Levin et al., 2011; Hammer and Levin, 2009) or for regional transport model comparison exercises (Taguchi et al., 2011).

The principal setup of the Heidelberg $^{222}$Rn Monitor (HRM) was designed in the 1990s (Levin et al., 2002). The monitor was modernized in 2010 by implementing state-of-the-art electronics as well as data acquisition and evaluation hardware and software (Rosenfeld, 2010). The system consists of a homemade filter holder, carrying a Whatman filter (QMA Ø 47mm), where all aerosols from an ambient airflow of ca. 1 m$^3$ per hour (measured with a Bronkhorst mass flow meter, model F-112AC-AAD-22-V) are actively deposited. A surface barrier detector (Canberra CAM 900 mm$^2$ active surface) mounted in

the filter holder at about 5 mm distance from the loaded filter surface measures the α-particles from the decaying $^{222}$Rn and $^{220}$Rn daughters. Hourly or half-hourly integrated α-spectra are stored for later separation of $^{214}$Polonium ($^{214}$Po) counts from the spectrum. Levin et al. (2002) explain evaluation of the spectra in detail. Taking into account the airflow rate through the filter, the filter efficiency (assumed as 100%) and the solid angle of the detector, which depends on the distance of the detector from the filter, finally allows estimating the atmospheric $^{214}$Po activity concentration. For calculation of ambient $^{222}$Rn activity

concentration, a correction for disequilibrium of $^{214}$Po has to be made (Levin et al., 2002; Schmithüsen et al., 2016).

### 2.2 Set-up of the line tests at the Heidelberg measurement site

To minimize aerosol loss in the sample intake, the standard intake line for the Heidelberg Radon Monitors (HRM) is a ca. 0.5 meter long 8 mm inner diameter Teflon tubing. In Heidelberg, this tubing is connected directly to outside air through a hole in the laboratory window. For investigating the effect of different intake lines on the measured $^{214}$Po activity concentration, we

used standard Decabon (SertoFlex) tubing with 12 mm outer and 8.2 mm inner diameter, which was varied in length between 16 m and 200 m. Before each test, the monitor with the long intake tube was run with a standard intake line for more than one week in parallel to our routine Heidelberg Radon monitor. This allowed us to determine a "calibration" of the so-called Line-Test-Monitor (LTM) relative to the Heidelberg routine instrument (HD-R). Small systematic differences of typically less than 5% can occur between monitors, which may be due to slightly different solid angles or efficiencies of the surface barrier

detectors or of the mass flow meter calibrations. From this comparison we estimate a statistical measurement uncertainty of activity ratios of two Heidelberg monitors of about 25% for ambient activity concentrations below 1 Bq m$^{-3}$ and of 10% or less for concentrations above 1 Bq m$^{-3}$ After the initial calibration period, each line test was run for at least three weeks to





make sure that the typical activity concentration range in Heidelberg of 1 – 15 Bq m$^{-3}$ was covered by the measurements. For these laboratory tests, the extended intake lines were rolled up to a diameter of approximately 0.5 m; they were always kept horizontally and at laboratory temperature of 19-22°C.

**2.3 Set-up of the Cabauw experiment (NL, 51°58'N 4°56'E, -0.7 m a.s.l.)**

Cabauw tower is a 213 m high tower, specifically built for meteorological research to establish relations between the states of the atmospheric boundary layer, land surface conditions and the general weather situation for all seasons. Cabauw tower is located in the western part of the Netherlands in a polder 0.7 m below average sea level. The North Sea is more than 50 km away to the WNW.

$^{222}$Rn is continuously measured at Cabauw station from two different heights, 20 m and 200 m above ground level; air is drawn at approximately 100 l min$^{-1}$ through 7 cm outer diameter terylene fiber water pipes through stack blowers into the measurement devices. These ANSTO radon monitoring devices are based on the two-filter technique (Whittlestone and Zahorowski, 1998; Zahorowski et al., 2004). The details of the Cabauw monitors and their calibration are described in Popa et al. (2011) as well as in the companion paper by Schmithüsen et al. (2016).

Our first set of measurements with a HRM at Cabauw tower (December 15, 2011 to February 21, 2012) was through a pre-installed Decabon tubing (in total 200 m) from the 180 m level down to the basement, where the HRM filter head was installed. This tubing had a slightly larger inner diameter than the tubing tested in the laboratory, namely 10 mm. A second set of measurements was performed with the HRM standard intake line (0.5 m) also from the 180 m platform (July 10 – August 26,
2012). For these measurements we assume no aerosol loss during air intake of the HRM; the respective data thus serve as reference for the compatibility of the two measurement systems (ANSTO and HRM) located at slightly different heights above ground (ANSTO: 200 m and HRM: 180 m). They allow to determine possible combined calibration biases or mean differences due to disequilibrium of the HRM-measured $^{214}$Po to the ANSTO-measured $^{222}$Rn activity concentrations (see accompanying paper by Schmithüsen et al., 2016).

**3 Results**

**3.1 Results of the line tests performed in Heidelberg**

Figure 1 shows an example of results from a line test with 8.2 mm inner diameter Decabon tubing in the Heidelberg laboratory. Here we plot the $^{214}$Po activity concentration measured on ambient air collected via a 200 m intake line (Line Test Monitor, LTM) versus that measured with the Heidelberg Routine (HD-R) monitor sampling the same air via a 0.5 m intake tubing. The
correlation between the two data sets is very good; however, the slope of the regression line is only 0.66, indicating that we lose more than 30% of $^{222}$Rn progeny activity concentration in the tubing. When plotting the ratio of the respective data versus





the activity measured with the LTM we can further see that at activity concentrations smaller than $\approx 2$ Bq m$^{-3}$, the relative activity loss for the LTM appears to be even larger than at high activities. Line tests with shorter tubing than 200 m show a similar pattern, however, the shorter the tubing the higher is the "saturation ratio" R at high atmospheric activity concentrations. The activity dependency of the ratios LTM/HD-R (e.g. Fig. 1b) can be approximated by a saturation curve according to the

following function

$$\frac{c_{LTM}}{c_{HD-R}} = R(L) - A \cdot \exp^{-\frac{c_{LTM}}{c_0}}.$$    (1)

The activity ratios from the 200 m line test in Fig. 1b have thus been fitted accordingly with the fit curve plotted as solid red

line. The parameter R(L) (dimensionless activity ratio at high concentrations) in Eq. (1) then corresponds to the saturation ratio, here ca. 0.66. Similar fit curves as shown in Fig. 1 have been calculated through all laboratory line test results with tubing between 16 m and 200 m. They yielded saturation ratios R(L) decreasing with increasing line length L; respective results are displayed in Fig. 2a, where the saturation ratios R(L) are plotted versus the length of the tubing of the LTM. An exponentially decreasing saturation value R(L) is observed with increasing length L of the tubing and a respective exponential fit is plotted

through the data (solid red line in Fig. 2a):

$$R(L) = \exp^{-\frac{L}{L_0}}.$$    (2)

The error bars in Fig. 2a correspond to the standard deviation of all measurements from each line test with activity

concentrations >7 Bq m$^{-3}$. Equation (2) could then principally be used to correct measurements for the loss of $^{222}$Rn progeny activity concentration at activity concentrations > 7 Bq m$^{-3}$ in 8.2 mm Decabon tubing.

In order to additionally account for the activity-dependent loss (i.e. at activity values $< \approx 7$ Bq m$^{-3}$), we corrected all our line test data with different line lengths for the general "saturation" loss and combined them into one single "normalized" data set

of corrected activity ratios (Fig. 2b). We then applied a joint fit through all data according to Eq. (1). This fit using the length-corrected data – by definition - yields a mean saturation value R = 1. The maximum activity concentration loss at very low activities ($c_{LTM} \rightarrow 0$) approaches 40% (Fig. 2b). The parameters of the joint fit as well as those of the fit for the general length dependent loss correction (as shown in Fig. 2b) are $L_0 = 415$ m; A = 0.41; $c_0 = 0.92$ Bq m$^{-3}$.

### 3.2 Evaluation of the empirical activity loss correction

In order to test if our empirical correction method yields reliable results, we have applied Eqs. (1 and 2), using the fit parameters from Fig. 2 to all individual line tests and plotted the results in Figs. S1 – S9 in the Supplement. These figures show in their upper left panels the original $c_{LTM}/c_{HD-R}$ ratios, in their lower left panels the ratios $c_{LTM}/c_{HD-R}$ of the LTM-corrected data, in





their upper right panels the linear correlation of corrected $c_{LTM}$ with $c_{HD-R}$ and in their lower right panels the regular residuum from the fit line in the lower left panel. For all line lengths the slopes lie between 0.997 and 1.064 and the scatter of the corrected ratios for activities >2 Bq m⁻³ is less than 20% in all cases, but with tubing longer than 50 m it may increase to more than 50% for activity concentrations lower than 1 Bq m⁻³. For a quantitative estimate of the average uncertainty of our

correction of the laboratory results, we display in Fig. 2c the ratios between normalized and corrected LTM data and the routine HRM data. In Fig. 2d, we calculated standard deviations for binned activity concentration ranges of 0.5 Bq m⁻³ each. The standard deviation is of order 60% for the lowest activity concentration range of 0 – 0.5 Bq m⁻³, but quickly decreases to less than 20% above 1 Bq m⁻³. Taking into account the uncertainty of activity ratios from two HRMs without tubing of ca. 25% below 1 Bq m⁻³ (see Section 2.2) and about 10% above 1 Bq m⁻³, the additional uncertainty of the correction for the effects of

long tubing is of order 40% below 1 Bq m⁻³, decreases to about 17% for 1.5 Bq m⁻³ and to less than 10% above 2 Bq m⁻³. We find a small bias in the corrected mean ratios of ca. 10% below 1.5 Bq m⁻³. A more sophisticated correction may allow for adjusting for that bias; however, we decided to keep the correction simple and valid for all tube lengths. In view of the relatively large statistical uncertainty of the measurement through long tubing (> 50 m) at activity concentrations below 1.5 Bq m⁻³ and in view of the fact that we cannot fully transfer our laboratory experiments to real field measurements (see below), we refrain

here from further second order adjustments.

**3.3 ²²²Rn progeny loss during ambient air sampling at Cabauw**

In addition to the line tests made with rolled-up 8.2 mm ID Decabon tubing in the Heidelberg laboratory and to evaluate the potential correction for radon progeny loss in vertical tubing, an additional line test has been made during the comparison campaign at Cabauw (Schmithüsen et al., 2016). From a direct comparison between the local ANSTO system, measuring ²²²Rn

at 200 m and the HRM, sampling air through a 0.5 m tubing at the 180 m platform, we obtained a mean difference between the ANSTO ²²²Rn data and the HRM ²¹⁴Po data of ANSTO/HRM = 1.11. The 11% bias between the two systems may be due to calibration and/or disequilibrium differences between the two systems, but is, most probably, due to calibration differences (Schmithüsen et al., 2016). For the line test we sampled air for the Heidelberg radon monitor (HRM) over two months through 200 m and 10 mm ID Decabon tubing from the 180 m level of the Cabauw tower, while the routine ANSTO instrumentation

was also measuring ²²²Rn in air collected at 200 m. In the comparison shown in Fig. 3, we have corrected the ANSTO data with the above mentioned calibration factor of 1/1.11, but there is still a significant difference visible between the two data sets. The slope of the regression line shown in Fig. 3b is HRM/ANSTO_corr = 0.73, indicating a mean loss of 27% of aerosol-bound ²²²Rn progeny in the 200 m tubing of the HRM. From our laboratory tests we would have expected a larger loss, i.e. 34% for a 200 m tubing. Note, however, that a tubing with slightly smaller inner diameter of 8.2 mm was used in the laboratory

tests, while at Cabauw tower, the tubing had an inner diameter of 10 mm, which can explain part of the difference. Moreover, in the laboratory tests the tubing was rolled up and lying on the floor, with potential loss due to inertial deposition on the bended tube walls as well as gravitational settling of aerosol and associated loss (see Discussion).



## 4 Discussion

The relative loss of $^{222}$Rn progeny activity concentrations (see Eq. 2: R(L)=A$_{line}$/A$_{direct}$) when sampling through a long tubing shows the expected correlation with tubing length L (assumed here to be exponential, but a linear correlation would also fit the measurement data) and a rather subtle dependency on the atmospheric activity concentration itself (Fig. 2b). Intake lines

are known to be a potential sink for aerosols (see e.g. Willeke and Baron (2005) and von der Weiden et al. (2009)), and all radon daughters measured with the one-filter HRM are attached to aerosols. The exact dependency of R(L) on the intake line length depends on the size of the aerosols carrying the radon daughter activity and on the flow regime in the tubing as well as its shape and orientation. Reineking et al. (1988) and Porstendörfer et al. (1994) report that most radon progeny activity in open air is carried by particles in the accumulation mode, with an average median aerodynamic diameter of 369 nm (range of

173 – 645 nm). Depending on meteorological conditions, a small fraction could also be found in the nucleation mode, with a diameter size range of 10-100 nm (Reineking et al., 1988). The flow regime in our Decabon tubing with d = 8.2 or 10 mm ID, a flow rate Q of about 1 m$^3$ hour$^{-1}$ and air (and particle) velocities between $3.5 < U < 5.3$ m s$^{-1}$, can neither be characterized as purely laminar (Reynolds number $Re < 2000$) nor as purely turbulent ($Re > 4000$). With $Re = U{\cdot}d/\nu$, where $\nu$ is the kinematic viscosity of the medium (in our case air at approximately 20°C, i.e. $\nu = 1.5\ 10^{-5}$ m$^2$s$^{-1}$), our Reynold numbers lie between 2350

and 2900, i.e. in the transitional regime, where for many loss processes no formula is available (von der Weiden et al., 2009).

For the laboratory experiment, four transport loss processes are potentially relevant to explain the observed radon progeny loss: (1) diffusion by Brownian motion towards the walls; (2) turbulent inertial deposition, when the (larger) particles cannot follow turbulent streamlines and thus stick to the walls; (3) gravitational settling in the horizontal tube; and (4) inertial

deposition on the bended tube walls. For the Cabauw experiment, with vertical tubing over most of the intake line, only the first two processes are potentially of relevance.

(1) Loss by molecular diffusion following a gradient towards the walls is stated by von der Weiden et al. (2009) as being relevant only for very small particles (<100 nm aerodynamic diameter). With aerodynamic diameters between 173 and 645 nm,

this process is probably negligible for the Heidelberg lab experiments and at Cabauw tower, too.

(2) Although our flow is in the transitional regime between laminar and turbulent, loss through turbulent inertial deposition may still be relevant. Von der Weiden et al. (2009) give an equation to estimate the turbulent inertial deposition efficiency (their Eq. 28), based on an experimentally determined turbulent inertial deposition velocity V$_t$ (Eq. 29) in tubing of length L

with diameter d

$$\eta_{turb,inert} = \exp(-\frac{\pi{\cdot}d{\cdot}L{\cdot}V_t}{Q}) \qquad (3)$$



$V_t$ can be estimated with Eq. 29 of von der Weiden et al. (2009), based on the Stokes number *Stk* and the Reynolds *Re* number according to

$$V_t = \frac{[6\cdot10^{-4}(0.0395\cdot Stk\cdot Re^{3/4})^2 + 2\cdot10^{-8}\cdot Re]\cdot U}{5.03\cdot Re^{1/8}}.$$

(4)

The Stokes number is calculated according to Brockmann (2005, his Eqs. 6-1 and 6-2) $Stk = V_{ts} \cdot U/(d \cdot g)$, where $V_{ts}$ is the gravitational settling velocity (in our case of aerosol with 369 nm median aerodynamic diameter estimated to about 2 cm h$^{-1}$ (Seinfeld and Pandis, 2006)), and g is the gravitational acceleration. With a Stokes number of $3.6 \cdot 10^{-4}$ for the lab experiments, Eq. (3) yields a transport efficiency of $\eta_{turb,inert} = 0.66$. For the Cabauw tower tubing of 10 mm ID the Stokes number is smaller,

i.e. $2.0 \cdot 10^{-4}$ and the aerosol loss by turbulent deposition is also smaller and estimated by Eqs. (3 and 4) to $\eta_{turb,inert} = 0.75$.

(3) In the laboratory tests, loss may also have been due to gravitational settling in the rolled-up tubes kept horizontally, although this effect is more important for larger particles >500 nm (von der Weiden et al., 2009). We can estimate the respective loss for *laminar* flow according to Eq. 24 of von der Weiden et al. (2009)

$$\eta_{tube,grav} = 1 - \frac{2}{\pi}\cdot\left(2\varepsilon\cdot\sqrt{1 - \varepsilon^{2/3}} - \varepsilon^{1/3}\cdot\sqrt{1 - \varepsilon^{2/3}} + arcsin\left(\varepsilon^{1/3}\right)\right),$$

(5a)

where $\varepsilon = \frac{3}{4} \cdot Z$ and $Z = L\cdot V_{ts}/(U\cdot d)$. For *turbulent* flow von der Weiden et al. (2009) give a different equation, i.e. their Eq. 26 (Eq. 6-40 in Brockmann (2005))


$$\eta_{tube,grav} = exp\left(-\frac{d\cdot L\cdot V_{ts}}{Q}\right),$$

(5b)

but for a settling velocity $V_{ts} \approx 2$ cm h$^{-1}$ (as estimated for our median diameter aerosol of 369 nm, see above) both equations (5a and 5b) yield the same loss, i.e. for a 200 m horizontal tube of d = 8.2 mm ID, a loss of about 3%. Loss due to gravitational

settling would double for a 645 nm median diameter aerosol and be only about 1% for 175 nm aerosol.

(4) The fourth process to be considered for loss in the laboratory experiment is due to inertial deposition at the walls of the bended tubing. Calculations of this lost process are commonly made following the data based empirical correction functions given by Pui et al. (1987) (see e.g. Brockmann (2005) and Von der Weiden et al. (2009)). For laminar flow, Pui et al. (1987)

show in their Figs. 7 and 8 that aerosol loss in bended tubing is negligible for Stokes numbers smaller than 0.01, which is the case in our laboratory experiments (see above). For turbulent flow, the latter authors developed a correction function independent of the Reynolds number and the curvature of the bend, i.e.



$$\eta_{bend,inert} = exp(-2.823\ Stk \cdot \phi), \tag{6}$$

where $\phi$ is the angle of the bend in radians and *Stk* is the Stokes number. Although the Stokes number in our experimental
setup for aerosol with median aerodynamic diameter of about 370 nm ($V_{ts} \approx 2$ cm h$^{-1}$) is small (ca. $3.6\ 10^{-4}$), in the laboratory
experiments with up to 200 m tubing rolled up, $\phi$ gets very large. With Eq. (6) we would thus obtain an efficiency of
$\eta_{bend,\ inert} = 0.45$, only for this loss process. The data set used by Pui et al. (1987) to establish Eq. 6 had, however, *Re* numbers
of 6000 and 10000, and a curvature ratio $R_0$ (i.e. the radius of the bend divided by the radius of the tube (Brockmann, 2005))
of $\sim 6$. Thus flow and bending conditions under which Eq. 6 was obtained were quite more turbulent and sharper bended,
respectively, compared to the conditions encountered within our experiment (Re = 2900 and $R_0 = 61$). But Pui et al. (1987)
affirm the insignificance of $R_0$ in Eq. 6 only for a range of $5 \leq R_0 \leq 30$. Considering this, we have to conclude that the conditions
of our experiment might be only partially or not at all suitable to be evaluated with Eq. 6, leaving a quantitative estimate of
this loss process open. Therefore, although we cannot fully rule out in our laboratory experiments aerosol loss due to inertial
deposition in the bended tubing, it is likely that this effect is only small. One strong argument for this assumption is that we
would have observed much larger differences in aerosol loss between laboratory experiments (with the rolled-up tubing) and
the field experiment at Cabauw tower, where $^{222}$Rn progeny were sampled through an almost straight vertical tubing.

In conclusion, we assume that the most relevant process for aerosol loss in both our experimental settings was most probably
through turbulent inertial deposition on the walls of the tubing. We favour this loss process also because, contrary to the other
loss processes, it does not depend much on the median aerodynamic diameter of the aerosol. In fact, we cannot expect the size
distributions of the $^{222}$Rn daughter-carrying aerosol to be identical at Cabauw tower located close to the sea and in Heidelberg,
a rather polluted and more continental station. A few percent of loss may have additionally occurred through gravitational
settling in the laboratory experiments.

The activity-dependency of aerosol loss, which was found in the laboratory (see Fig. 2b) as well as in the field experiment at
Cabauw (not shown), is more difficult to explain. One hypothesis is that at low activity concentration, also the aerosol
concentration is lower with a potentially shifted size distribution of the progeny-carrying aerosol towards smaller aerodynamic
diameters. Small particles have a higher diffusion coefficient than the median diameter aerosol and have thus a higher
probability to be lost by Brownian diffusion to the walls of the tubing. However, this explanation is currently only speculative.

**5 Conclusions**

Our laboratory and field experiments provided first results on $^{222}$Rn progeny loss in long Decabon tubing of 8.2 and 10 mm
inner diameter (ID). As the most probable loss mechanism, we identified turbulent inertial deposition on the walls of the tubing,





as the theoretical estimates of this loss process would equally well explain our findings in the two different experimental settings, i.e. in the laboratory with horizontally kept rolled-up 8.2 mm ID tubing and in the field with 10 mm ID straight vertical tubing. In these two cases, we estimate sampling efficiencies for 200 m long tubing between 0.66 (8.2 mm ID) and 0.75 (10 mm ID). When also considering for the rolled-up tubing kept horizontally in the laboratory additional loss due to gravitational

settling, the total sampling efficiency is reduced to 0.64. These theoretical estimates are in surprisingly good agreement with our experimental data that yielded efficiencies of 0.66 for the rolled-up tubing in the laboratory and 0.73 at the Cabauw tower.

Based on this agreement, the correction function, which was empirically derived from our laboratory experiments, showing an exponential increase of loss with tube length seems adequate and applicable for real field measurements. Likewise, one may

take the theoretical approach (Eq. 3 and 4) to correct for $^{222}$Rn progeny loss in the tubing. As the turbulent inertial loss is not sensitive to the median aerodynamic diameter of the aerosol, the theoretical approach may even be the better choice for correction. The associated uncertainty of the correction is only a few percent.

The observed activity-dependency of the loss, when sampling through long tubing, could possibly be explained by decreasing

aerodynamic diameter of the radon progeny-carrying aerosol. In order to correct for this effect we suggest applying our empirical observation-based function given by Eq. 1. However, one should keep in mind that the uncertainty of this correction is strongly increasing with decreasing atmospheric activity concentration. An additional uncertainty of 10-20% will be associated with the corrected data, if $^{214}$Po activity concentrations lie between 1 and 2 Bq m$^{-3}$. For activity concentrations above 2 Bq m$^{-3}$, the additional uncertainty of the correction is smaller than 5%, which seems well acceptable in view of other

uncertainties associated with the $^{222}$Rn progeny measurement. But for activity concentrations below 1 Bq m$^{-3}$, which very often occur at coastal stations or at sites not too far from the sea, like Cabauw (see Fig. 3), $^{222}$Rn progeny sampling through long tubing (i.e. longer than 50 m) is not recommended. Here our proposed correction will have huge uncertainty (>50%, see also Supplementary Figures). For more continental stations, i.e. more than a few hundred kilometers from the coastline, $^{222}$Rn activity concentrations drop below 1 Bq m$^{-3}$ only during intensive vertical mixing during summer days or when the air masses

have stayed over the continent for less than a day and the continental $^{222}$Rn pile-up is still low. Therefore, at these stations one may well accept additional statistical uncertainty due to line loss corrections, also of order 10%, which would be in the same range as the uncertainty of disequilibrium between $^{222}$Rn and its measured progeny (see accompanying paper by Schmithüsen et al. (2016)).

*Acknowledgements:* The research leading to these results has received funding from the European Community's Seventh Framework Program (FP7/2007-2013) in the InGOS project under grant agreement No. 284274. We wish to thank the technicians at Cabauw station, Pim van den Bulk and Mark Blom, for their support during the comparison study and Susanne Preunkert, LGGE, Grenoble for helpful discussions.





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





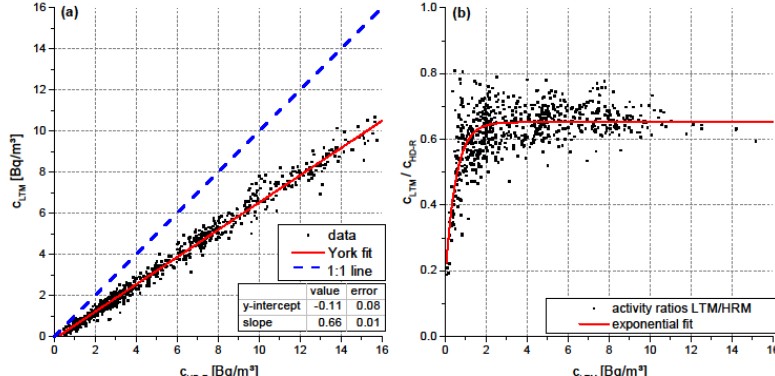

**Figure 1. a: Correlation of the $^{214}$Po activity concentration measured with the line test monitor (LTM) sampling via a 200 m intake line versus that measured with the Heidelberg routine monitor (HD-R). The red line is a least squares fit (York et al., 2004) through the data yielding a slope of 0.66; the dashed blue line shows the 1:1 relation. b: Activity concentration ratio of the LTM and the HD-R monitor plotted vs. the activity concentration measured with the LTM. The red line is an exponential fit through the data according to Eq. (1).**





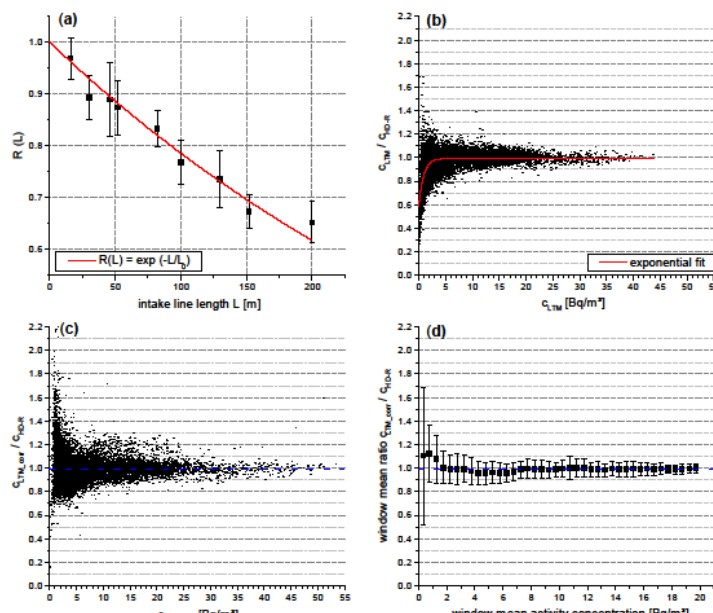

**Figure 2 (a): Length-dependent saturation values as determined from individual line tests (compare also Supplementary Figs. S1 – S9). The solid line is an exponential fit calculated through all data according to Eq. (2). (b): Length-corrected line test results (i.e.**
5 **normalized to R=1) together with the saturation fit curve calculated according to Eq. (1). (c) Activity concentration ratio of corrected LTM data and HD-R data without tubing. Panel (d) gives the standard deviations of ratios displayed in (c) binned into 0.5 Bq m$^{-3}$ ranges.**



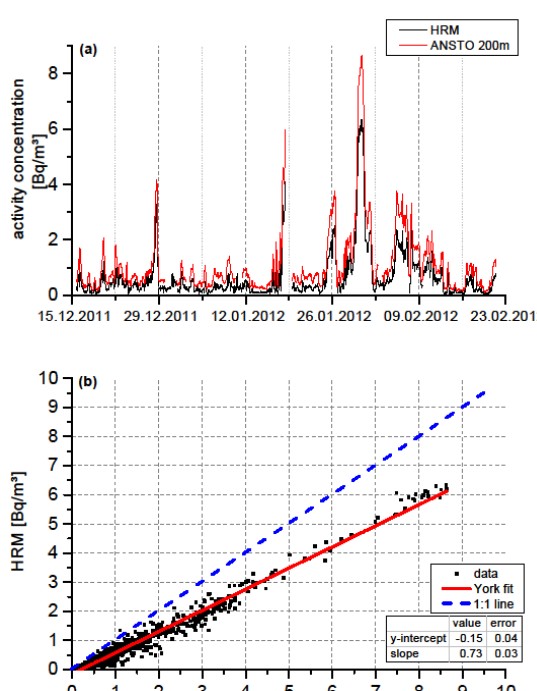

**Figure 3: Results from the line test experiment at Cabauw tower. The upper panel (a) shows ²²²Rn activity concentrations measured with the ANSTO system on air collected from the 200 m platform (red line), but corrected by a factor of 1/1.11 (see text) in comparison to the HRM ²¹⁴Po measurements on air from the 180 m level of the tower collected through a 200 m Decabon tubing (black line). The lower panel (b) shows the scatter plot of the two records with a geometric mean regression line (York, et al., 2004) yielding a slope of 0.73.**