# Peer review of "Assessment of 222Radon progeny loss in long tubing based on static filter measurements in the laboratory and in the field"

_Atmospheric Measurement Techniques, 2016_

## Referee Comment (RC1) · Anonymous Referee #1 · 6 Jul 2016

General comments

A very interesting and useful contribution on the consequences of using long sampling lines in single-filter atmospheric radon measurements. A simple method is developed, and verified both in the laboratory and field, for correcting single-filter radon measurements for the effects of losses due to deposition of aerosol-attached radon progeny onto the walls of inlet tubes of varying lengths up to 200m. The results are discussed with reference to a number of possible aerosol deposition mechanisms inside tubing, with the outcome that the technique is both supported theoretically and has its limitations specifically identified and quantified. The prospect that inlet tubing deposition effects can be accurately corrected will reduce an important source of uncertainty in

single-filter radon measurements, and expand the list of potential applications for these (relatively compact) detectors to include tall tower sites. This manuscript falls within the scope of AMT and describes topical original research worthy of publication. Furthermore, the manuscript is for the most part clearly written and well structured. I therefore recommend acceptance, after attention has been paid to a few minor issues and technical corrections listed below.

Minor changes / technical corrections

Abstract

P1, L11: Change "tubing (laboratory test) and" to "tubing in the laboratory, and".

P1, L12: "Progeny loss increased exponentially with length of the tubing". I think this is over-stating it a bit, as the increase could equally well be described as "linear". Perhaps just say "Progeny loss increased steeply with length of the tubing".

P1, L13: After "for 8.2 mm ID rolled-up tubing of 200 m length", I think you should add "at a flow rate of 1 mˆ3 hourˆ-1" or similar. I think it is important to specify the flow rate, as the deposition efficiency is strongly dependant on it (see Eq. 3).

P1, L19: Change "above 1" to "between 1-2".

1 Introduction

P1, L22: Change "used as tracer" to "used as a tracer".

P1, L25-26: Change "shortly after the generation of the progeny" to "shortly after generation in the atmosphere".

P1, L26-27: Rewrite sentence as "In so-called two-filter systems, filtered air that only contains 222Rn (with ambient progeny removed) is flushed through a large delay chamber and the new 222Rn progeny produced in situ inside the tank are then collected on a second internal filter. . .".

P1, L30: "In this style of detector, atmospheric 222Rn activity concentration is then. . .".

P2, L8: Change "implemented e.g. at tower stations, where" to "implemented at locations (e.g. tower stations) where".

P2, L22: Change "Zahorowski, 1998). Alternatively, the Heidelberg" to "Zahorowski, 1998), whereas the Heidelberg".

P2, L23-24: Delete ", and routine maintenance work and data evaluation are less demanding than for the two-filter systems (Whittlestone and Zahorowski, 1998)". This is not generally true with modern versions of the two-filter detectors, which are very robust and include automated calibration and background checking systems that allow them to operate for months or even years at a time without human intervention.

P2, L32: Change "Whittelstone" to "Whittlestone".

2 Methods

P3, L7: Change "used as tracer" to "used as a tracer".

P3, L19: Change "estimating the atmospheric" to "estimation of the atmospheric".

P3, L31: Change "ratios of two" to "ratios for the two".

P4, L3: It would be nice to know if there are any ambient temperature or humidity effects on deposition rates, but I guess this was out of the scope of your study. However, I would have thought that it would be fairly easy to test the effects of different flow rates, especially given the expected strong dependency on Q (see Eq. 3). Did you investigate flow rate sensitivity?

P4, L20: Change "For these measurements" to "For these latter measurements".

P4, L22: Change "They allow to determine possible" to "They allow the determination of possible".

P4, L24: How is the flow rate maintained in the 200m tube on the tower? Was it the

same as in the laboratory tests? Given the expected strong dependency of deposition efficiency on Q (Eq. 3), it would have been good to know the variability of flow rate achieved through the 200m tube on the tower.

3 Results

P5, L9-11: Need to rearrange / change the first two sentences after Eq. (1) in order to explain some of the terms better. I suggest something like: "where parameters A and C0 are constants, and R(L) is the "saturation" activity ratio for line length L. The activity ratios for the 200 m line test have been fitted accordingly in Fig. 1b, with the fit curve plotted as a solid red line with R(L)=0.66".

P5, L12: Change "They yielded" to "This process yielded".

P5, L13: "An exponentially decreasing...". Looking at the plot, this could equally well be described by a linear fit. Maybe you should explain your choice of an exponential fitting function (e.g. we would expect R(L)→ 0 as L→∞).

P5, L19: Define L0 (e.g. "where L0 is a constant parameter").

P5, L20: Change "then principally" to "therefore in principle".

P5, L23-25: Please elaborate briefly how you corrected and "normalised"? I guess you added [1-R(L)] to both sides of Eq. (1), which is equivalent to modifying CLTM on the LHS by adding CHD-R[1-R(L)]?

P5, L28: Change "as shown in Fig. 2b" to "as shown in Fig. 2a".

P6, L24: At what flow rate? What was the variability of the flow rate?

4 Discussion

P9, L9: Change "were quite more turbulent and sharper bended" to "were quite a lot more turbulent and more sharply bended".

5 Conclusions

P10, L25: Change "pile-up" to "accumulation".

---

## Referee Comment (RC2) · Anonymous Referee #2 · 10 Jul 2016

Comments on "Assessment of $^{222}$Radon progeny loss in long tubing based on static filter measurements in the laboratory and in the field"

Referee #2

9 July 2016

GENERAL COMMENTS

This paper deals with measurement of atmospheric Po214, and indirectly atmospheric Rn222, by allowing the sampled air to first pass through long lengths of tubing before direct measurement of the alpha decays of Po214 deposited on an exit filter. If a predictable relation between Po214 entering the tubing and that present at the exit filter can be established, simplification in the measurement of atmospheric radon at the top of high meteorological towers might be possible. Much of the labor intensive equipment could be located at the bottom of the towers with only a tubing extension to the less accessible elevated intake sites above. The authors make experimental measurements using tubing lengths about 10 mm inside diameter and up to 200 m in length and determine the fraction of entering Po214 present at the exit end. They compare their measurements with predictions of a single-mode-size aerosol transport model and get good agreement except at the lowest Po214 concentrations.

There is good reason why over the years people designing atmospheric measurement systems for radon and its progeny have chosen to use short, wide intake tubes and high flow rates. Through hard experience they have learned that losses of progeny passing through tubing or chambers are difficult to predict or correct for. There are just too many factors, some uncontrollable, that can affect losses of radon progeny in transport: turbulence levels, humidity, temperature, aerosol concentration, aerosol size distribution, electric fields, and tubing material, to name a few. So among traditional designers of precision atmospheric measurement systems it would be almost heresy to consider using a 200 m diameter intake tube if it could be at all avoided.

The above said, I think the present authors provide some valid arguments for considering long intake tubing. I also think their experimental measurements are sound and results reasonable. However, one limitation of their measurements is that they apparently do not report concurrent measurements of aerosol size and concentration at the intake. Other concurrent measurements might also be useful, such as temperature and humidity. Such measurements might prove useful for understanding the model's difficulties at low concentrations, and, more importantly, for projecting results to other monitoring sites where atmospheric conditions might be much different.

Although their modeling effort seemed valid given its approximations, I found it less impressive. It is just too simplified to properly capture some of the important nuances involved. This is actually a very difficult fluid dynamics mass transport problem that cannot be solved exactly. However, more refined modelling is possible. In principle, I would do something such as the following. Specify the concentration of radon, and radon progeny (in BOTH) the attached to aerosol and unattached to aerosol modes at the entrance to the tubing. Then, do a time-dependent calculation of a parcel of air as it passes along the tubing, keeping track of the important sinks and sources for Po214, such as deposition

of attached and, separately, unattached (characterized by effective deposition velocities), decay, ingrowth, etc.  I think such a more refined model might have a better chance of coming up with an explanation for the failure of their model at low concentrations.

So let me summarize my evaluation of this paper.  It has a core of solid new experimental results.  These results are not surprising but might be valuable for others considering the long tubing approach. Comparison with estimates of a simple model is worth discussing.  However, the paper would be stronger if it had more information about air conditions at the entrance to the tubing.  It would also be stronger if more refined modeling were carried out.  Both of these revisions would be helpful if readers hope to reliably project the present results to other measurement sites with different atmospheric conditions and with different measurement equipment.  If the authors choose not to carry out revisions of this type, they should then at least state more clearly the limited promise of the long tubing approach.  If we are talking about international standards and global monitoring of atmospheric radon at major meteorological sites, I would put the long tubing approach and one filter Po214 measurement far down the list of preferred or reliable techniques.  Data from these global stations have a way of working their way into data bases then used by unsuspecting modelers to draw important conclusions about atmospheric circulation, climate change, and global air pollution.

COMMENTS ON REFERENCES

The paper does a fairly good job of citing relevant references.  The cited papers by Porstendorfer (1994) and Von der Weiden (2009) are particularly relevant. However, if they have not already done so, the authors might take a look at the following references that could provide additional relevant information.

1)Beyond Porstendorfer's helpful discussion, there are potentially other factors that can come into play controlling the physical behavior of radon progeny in air spaces.  Although it is an old paper dealing with atmospheric radon progeny in a different context, and a bit of an overkill, the paper by A. Roffman, Journal of Geophysical Research, vol. 77, #10, 1972, 5883-5899, is useful at giving an idea of the many factors that have to be considered in a modelling transport of atmospheric radon progeny.

2) There is an important update to the paper by Jacobi and Andre (1963) that refines their modelling by using two progeny-size groups (attached and unattached) and more realistic boundary conditions at the earth's surface.  This class of model is much better at predicting and explaining disequilibrium of radon progeny near the earth's surface in the zone relevant for meteorological towers.  It more clearly brings out the important role of the atmospheric aerosol concentration.  See Schery and Wasiolek, Journal of Geophysical Research, vol. 98, #D12, 1993, 22915 – 22923, and references therein.

3)  I'm not an expert on all the issues related to making reference-grade measurements of atmospheric radon at meteorological towers, but, as a starting point, would take a look at R. Colle et al, Journal of Geophysical Research: Atmospheres, Vol. 100, Issue D8, 1995, pages16617 to 16638, and references therein.  References in this paper should provide leads to previous work dealing with precision measurement of atmospheric and correcting for losses of radon progeny on the way to an exit filter.

PAGE BY PAGE COMMENTS

Page 3 and 4. Set-up sections and elsewhere. I would specifically make clear flow rates used and time of transit along tubing. These are key factors controlling the amount of progeny lost during passage along the tubing. It would also be important to monitor aerosol concentration, and even size distribution, at the entrance to the tubing. If I read the paper correctly, such aerosol measurements are not reported; instead, a generic estimate for typical conditions is given. If not available, such concurrent aerosol monitoring should be considered in future studies. Aerosol information, flow rates, tubing diameter, and transit times are all important for controlling losses of progeny along the tubing. Any person wishing to replicate the present results at another site would need to know this information.

Page 5. Equation 1. Here and elsewhere, make sure all variables and constants are clearly explained when first mentioned. In this case, are $c_0$ and A free parameters adjusted to achieve an optimum prediction of the data?

Page 6. "an additional line test" >>> "a line test". The word "additional" has already been stated.

Page 7. First paragraph. This information on aerosol concentration and the fraction present in the nuclei mode is important for understanding the paper's results. As far as I can see, the authors are not reporting their own measurements concurrent with the Po214 measurements. If I am correct, this is a weakness of this paper.

Page 7 and page 8. The discussion of four possible loss processes. I think it is probably true that losses due to the physical behavior of the accumulation mode size aerosol are important and worth discussing in the spirit of a "first order estimate". However, beyond that, the situation is probably much more complex, particularly at different flow rates and with other possible, but less common for the sites studied, intake conditions such as "clean air" and a high intake fraction of unattached progeny. Ideally, redo the modelling using a two mode model. Otherwise, maybe shorten this section making clear the problem is too difficult to warrant more than a first order estimate in an experimentally oriented paper.

Page 10. "The observed activity-dependency of the loss . . ." I suspect this faithfully measured and reported variation in the transmission efficiency is only the tip of the iceberg. The unexplained variation at low concentrations is probably due to something like clean air with lower aerosol concentrations and/or a higher unattached fraction of progeny at the tubing intake -- conditions more common at certain other meteorological towers. The present modelling and experimental protocol are inadequate to identify the exact cause, much less provide other researchers with the tools to adequately predict losses at a different site, with different meteorological conditions, with different flow rates, and with different tubing. This is why I recommend the following. If the present authors have the ability to make some the experimental and modelling revisions I suggest, that would strengthen the paper. Otherwise, I think their conclusions should be more carefully stated along the lines of "… We have made careful measurements for one particular set of test conditions and analyzed our results with a first order model. Given the significant progeny losses we observed with long sampling tubing, and unexplained variations

in the loss fraction, we conclude that the approach of using long tubing intakes to measure atmospheric radon and Po214 is not presently reliable enough to recommend for situations where reference-grade measurements of atmospheric Rn are required." In addition, with a paper of more limited scope, it might be possible to shorten the paper and/or move more material to the supplement.

---

## Author Comment (AC1) · 5 Oct 2016

Replies to reviewer comments #1

We wish to thank the reviewer for his/her valuable comments on the manuscript and suggestions for changes that largely helped to improve it. Our replies to the individual remarks are printed in blue color:

A very interesting and useful contribution on the consequences of using long sampling lines in single-filter atmospheric radon measurements. A simple method is developed, and verified both in the laboratory and field, for correcting single-filter radon measurements for the effects of losses due to deposition of aerosol-attached radon progeny onto the walls of inlet tubes of varying lengths up to 200m. The results are discussed with reference to a number of possible aerosol deposition mechanisms inside tubing, with the outcome that the technique is both supported theoretically and has its limitations specifically identified and quantified. The prospect that inlet tubing deposition effects can be accurately corrected will reduce an important source of uncertainty in single-filter radon measurements, and expand the list of potential applications for these (relatively compact) detectors to include tall tower sites. This manuscript falls within the scope of AMT and describes topical original research worthy of publication. Furthermore, the manuscript is for the most part clearly written and well structured. I therefore recommend acceptance, after attention has been paid to a few minor issues and technical corrections listed below.

Minor changes / technical corrections

Abstract

P1, L11: Change "tubing (laboratory test) and" to "tubing in the laboratory, and".

Done

P1, L12: "Progeny loss increased exponentially with length of the tubing". I think this is over-stating it a bit, as the increase could equally well be described as "linear". Perhaps just say "Progeny loss increased steeply with length of the tubing".

OK, true, we changed it.

P1, L13: After "for 8.2 mm ID rolled-up tubing of 200 m length", I think you should add "at a flow rate of 1 mˆ3 hourˆ-1" or similar. I think it is important to specify the flow rate, as the deposition efficiency is strongly dependent on it (see Eq. 3).

Yes, has been changed

P1, L19: Change "above 1" to "between 1-2".

Done.

1 Introduction

P1, L22: Change "used as tracer" to "used as a tracer".

Done

P1, L25-26: Change "shortly after the generation of the progeny" to "shortly after generation in the atmosphere".

Done

P1, L26-27: Rewrite sentence as "In so-called two-filter systems, filtered air that only contains 222Rn (with ambient progeny removed) is flushed through a large delay chamber and the new 222Rn progeny produced in situ inside the tank are then collected on a second internal filter. . .".

Done, thank you.

P1, L30: "In this style of detector, atmospheric 222Rn activity concentration is then. . .".

Done

P2, L8: Change "implemented e.g. at tower stations, where" to "implemented at locations (e.g. tower stations) where".

We changed that.

P2, L22: Change "Zahorowski, 1998). Alternatively, the Heidelberg" to "Zahorowski, 1998), whereas the Heidelberg".

Done

P2, L23-24: Delete ", and routine maintenance work and data evaluation are less demanding than for the two-filter systems (Whittlestone and Zahorowski, 1998)". This is not generally true with modern versions of the two-filter detectors, which are very robust and include automated calibration and background checking systems that allow them to operate for months or even years at a time without human intervention.

We removed the second part of the sentence.

P2, L32: Change "Whittelstone" to "Whittlestone".

Corrected

2 Methods

P3, L7: Change "used as tracer" to "used as a tracer".

Changed

P3, L19: Change "estimating the atmospheric" to "estimation of the atmospheric".

Changed

P3, L31: Change "ratios of two" to "ratios for the two".

Changed

P4, L3: It would be nice to know if there are any ambient temperature or humidity effects on deposition rates, but I guess this was out of the scope of your study. However, I would have thought that it would be fairly easy to test the effects of different flow rates, especially given the expected strong dependency on Q (see Eq. 3). Did you investigate flow rate sensitivity?

Unfortunately, we did not test experimentally the dependency of flow rate on aerosol loss in the tubing, and you are correct about the rather strong dependency of the loss, based on Equation 3 and 4, (much less though on Eqs. 5a and b). Based on these four equations, we have now estimated the effect of flow rate differences from 1 m3 hour-1 in our laboratory experiments with the different line lengths (and pumps used). The maximum theoretical deviation of loss (6.8%) occurred at a flow rate

of 1.4 $m^3$ $hour^{-1}$ with the 100m line. Note that a flow rate of 1 $m^3$ $hour^{-1}$ (±10%) was used in all experiments with lines longer than 100m. To take this effect into account for the general loss correction (Eq. 2), we have now added in Figure 2 corrected values that have been adjusted to a nominal flow rate of 1 $m^3$ $hour^{-1}$ as well as a second fit curve through these corrected values. As all flow rates for line lengths ≤ 100 m were higher than 1 $m^3$ $hour^{-1}$, all these adjustment are positive and the characteristic length $L_0$ of the fit curve is about 10% larger ($L_0$ = 459 versus 415). When using the adjusted fit curve for R(L) this would lead to a maximum difference in the loss correction of 4.5% with a 200m line. The revised Figure 2a is displayed below.

[Figure]

P4, L20: Change "For these measurements" to "For these latter measurements".

Changed

P4, L22: Change "They allow to determine possible" to "They allow the determination of possible".

Changed

P4, L24: How is the flow rate maintained in the 200m tube on the tower? Was it the same as in the laboratory tests? Given the expected strong dependency of deposition efficiency on Q (Eq. 3), it would have been good to know the variability of flow rate achieved through the 200m tube on the tower.

The mean flow rate was here 1.1 $m^3$ $hour^{-1}$ (±10%), i.e. similar to the laboratory line tests with tubing > 100m. Theoretically we estimate a 2% change of aerosol loss for a 10% change in flow rate.

3 Results

P5, L9-11: Need to rearrange / change the first two sentences after Eq. (1) in order to explain some of the terms better. I suggest something like: "where parameters A and C0 are constants, and R(L) is the "saturation" activity ratio for line length L. The activity ratios for the 200 m line test have been fitted accordingly in Fig. 1b, with the fit curve plotted as a solid red line with R(L)=0.66".

We changed this part accordingly, thank you.

P5, L12: Change "They yielded" to "This process yielded".

Changed

P5, L13: "An exponentially decreasing. . .". Looking at the plot, this could equally well be described by a linear fit. Maybe you should explain your choice of an exponential fitting function (e.g. we would expect R(L) → 0 as L → ∞).

We added a respective sentence.

P5, L19: Define L0 (e.g. "where L0 is a constant parameter").

Added

P5, L20: Change "then principally" to "therefore in principle".

Changed

P5, L23-25: Please elaborate briefly how you corrected and "normalised"? I guess you added [1-R(L)] to both sides of Eq. (1), which is equivalent to modifying CLTM on the LHS by adding CHD-R[1-R(L)]?

We divided all data by their respective saturation value, this was added to the text.

P5, L28: Change "as shown in Fig. 2b" to "as shown in Fig. 2a".

Changed, thanks.

P6, L24: At what flow rate? What was the variability of the flow rate?

The flow rate was 1.1 $m^3$ $hour^{-1}$ (±10%)

4 Discussion

P9, L9: Change "were quite more turbulent and sharper bended" to "were quite a lot more turbulent and more sharply bended".

Changed

5 Conclusions

P10, L25: Change "pile-up" to "accumulation"

Changed

---

## Author Comment (AC2) · 5 Oct 2016

We wish to thank the reviewer for his/her insightful comments on the manuscript. We have revised the manuscript according to the suggestions. Our replies to the individual remarks are printed in blue color:

GENERAL COMMENTS

This paper deals with measurement of atmospheric Po214, and indirectly atmospheric Rn222, by allowing the sampled air to first pass through long lengths of tubing before direct measurement of the alpha decays of Po214 deposited on an exit filter. If a predictable relation between Po214 entering the tubing and that present at the exit filter can be established, simplification in the measurement of atmospheric radon at the top of high meteorological towers might be possible. Much of the labor intensive equipment could be located at the bottom of the towers with only a tubing extension to the less accessible elevated intake sites above. The authors make experimental measurements using tubing lengths about 10 mm inside diameter and up to 200 m in length and determine the fraction of entering Po214 present at the exit end. They compare their measurements with predictions of a single-mode-size aerosol transport model and get good agreement except at the lowest Po214 concentrations.

There is good reason why over the years people designing atmospheric measurement systems for radon and its progeny have chosen to use short, wide intake tubes and high flow rates. Through hard experience they have learned that losses of progeny passing through tubing or chambers are difficult to predict or correct for. There are just too many factors, some uncontrollable, that can affect losses of radon progeny in transport: turbulence levels, humidity, temperature, aerosol concentration, aerosol size distribution, electric fields, and tubing material, to name a few. So among traditional designers of precision atmospheric measurement systems it would be almost heresy to consider using a 200 m diameter intake tube if it could be at all avoided.

We fully agree to the reviewer's above comment and "warning" and decided to follow his/her advice to make clear that the results and correction factors presented here are only valid for this particular setting, i.e. tubing diameter and material, flow rate, aerosol size distribution and observational stations. Honestly, we were rather surprised ourselves that the radon progeny losses were comparably minor and often not detectable if tubing of a few 10m lengths had been tested (i.e. of order 10% or less). We are aware that collecting air for radon progeny measurements through long tubing should be avoided. Still, we often face logistic circumstances where air sampling through (long) tubing is the only option. The aim of our study was therefore to quantify systematic biases caused by this compromise and estimate uncertainties if we decide to apply corrections.

The above said, I think the present authors provide some valid arguments for considering long intake tubing. I also think their experimental measurements are sound and results reasonable. However, one limitation of their measurements is that they apparently do not report concurrent measurements of aerosol size and concentration at the intake. Other concurrent measurements might also be useful, such as temperature and humidity. Such measurements might prove useful for understanding the model's difficulties at low concentrations, and, more importantly, for projecting results to other monitoring sites where atmospheric conditions might be much different.

We thank the reviewer for his/her valuable suggestions for an improved experimental setup. We are indeed planning a new comparison study at a meteorological tower close to Heidelberg with co-located direct $^{222}$Rn and $^{214}$Po measurements as well as with a second $^{214}$Po measurement where air sampling is through Decabon tubing. Results from this planned study will, however, not be available in the next months and could thus not be included in the present paper.

Although their modeling effort seemed valid given its approximations, I found it less impressive. It is just too simplified to properly capture some of the important nuances involved. This is actually a very difficult fluid dynamics mass transport problem that cannot be solved exactly. However, more refined modelling is possible. In principle, I would do something such as the following. Specify the concentration of radon, and radon progeny (in BOTH) the attached to aerosol and unattached to aerosol modes at the entrance to the tubing. Then, do a time-dependent calculation of a parcel of air as it passes along the tubing, keeping track of the important sinks and sources for Po214, such as deposition of attached and, separately, unattached (characterized by effective deposition velocities), decay, ingrowth, etc. I think such a more refined model might have a better chance of coming up with an explanation for the failure of their model at low concentrations.

We agree that the suggested approach could provide more realistic results. However, our modelling results were only meant to provide the order of magnitude for the different potential loss processes. We are simply lacking a number of required parameters, such as the attached and the un-attached fraction of radon progeny, so that an improved theoretical approach is just not possible.

So let me summarize my evaluation of this paper. It has a core of solid new experimental results. These results are not surprising but might be valuable for others considering the long tubing approach. Comparison with estimates of a simple model is worth discussing. However, the paper would be stronger if it had more information about air conditions at the entrance to the tubing. It would also be stronger if more refined modeling were carried out. Both of these revisions would be helpful if readers hope to reliably project the present results to other measurement sites with different atmospheric conditions and with different measurement equipment. If the authors choose not to carry out revisions of this type, they should then at least state more clearly the limited promise of the long tubing approach. If we are talking about international standards and global monitoring of atmospheric radon at major meteorological sites, I would put the long tubing approach and one filter Po214 measurement far down the list of preferred or reliable techniques. Data from these global stations have a way of working their way into data bases then used by unsuspecting modelers to draw important conclusions about atmospheric circulation, climate change, and global air pollution.

We chose the second option suggested by the reviewer, namely making clear in the revised manuscript that our experiments provide correction factors only for this particular setting (tubing type, flow rate, aerosol size distribution, air intake height, etc.) and de-emphasize the applicability of the theoretical loss calculations to other settings in the revised manuscript.

COMMENTS ON REFERENCES

The paper does a fairly good job of citing relevant references. The cited papers by Porstendorfer (1994) and Von der Weiden (2009) are particularly relevant. However, if they have not already done so, the authors might take a look at the following references that could provide additional relevant information.

1) Beyond Porstendorfer's helpful discussion, there are potentially other factors that can come into play controlling the physical behavior of radon progeny in air spaces. Although it is an old paper dealing with atmospheric radon progeny in a different context, and a bit of an overkill, the paper by A. Roffman, Journal of Geophysical Research, vol. 77, #10, 1972, 5883-5899, is useful at giving an idea of the many factors that have to be considered in a modelling transport of atmospheric radon progeny.

2) There is an important update to the paper by Jacobi and Andre (1963) that refines their modelling by using two progeny-size groups (attached and unattached) and more realistic boundary conditions

at the earth's surface. This class of model is much better at predicting and explaining disequilibrium of radon progeny near the earth's surface in the zone relevant for meteorological towers. It more clearly brings out the important role of the atmospheric aerosol concentration. See Schery and Wasiolek, Journal of Geophysical Research, vol. 98, #D12, 1993, 22915 – 22923, and references therein.

3) I'm not an expert on all the issues related to making reference-grade measurements of atmospheric radon at meteorological towers, but, as a starting point, would take a look at R. Colle et al, Journal of Geophysical Research: Atmospheres, Vol. 100, Issue D8, 1995, pages16617 to 16638, and references therein. References in this paper should provide leads to previous work dealing with precision measurement of atmospheric and correcting for losses of radon progeny on the way to an exit filter.

We agree that the work by Schery and Wasiolek from 1993 presents an improved model of radon progeny behavior near the earth's surface and include the reference in the revised manuscript. Still, we think that at continental European stations with aerosol concentrations $>10^4$ cm$^{-3}$ (and even at Cabauw close to the North Sea coast) and at intake heights above 30m the un-attached fraction is most probably less than a few percent. But we thank the reviewer for bringing this point up as the unattached (charged) fraction may indeed be larger at low ambient $^{214}$Po activity (and aerosol) concentrations, thus potentially explaining the increased loss in the tubing at low ambient $^{214}$Po activity concentrations.

PAGE BY PAGE COMMENTS

Page 3 and 4. Set-up sections and elsewhere. I would specifically make clear flow rates used and time of transit along tubing. These are key factors controlling the amount of progeny lost during passage along the tubing. It would also be important to monitor aerosol concentration, and even size distribution, at the entrance to the tubing. If I read the paper correctly, such aerosol measurements are not reported; instead, a generic estimate for typical conditions is given. If not available, such concurrent aerosol monitoring should be considered in future studies. Aerosol information, flow rates, tubing diameter, and transit times are all important for controlling losses of progeny along the tubing. Any person wishing to replicate the present results at another site would need to know this information.

We included the missing information, if available.

Page 5. Equation 1. Here and elsewhere, make sure all variables and constants are clearly explained when first mentioned. In this case, are c0 and A free parameters adjusted to achieve an optimum prediction of the data?

Yes, $C_0$ and A are free parameters, this is now mentioned in the revised manuscript.

Page 6. "an additional line test" >>> "a line test". The word "additional" has already been stated.

Has been changed.

Page 7. First paragraph. This information on aerosol concentration and the fraction present in the nuclei mode is important for understanding the paper's results. As far as I can see, the authors are not reporting their own measurements concurrent with the Po214 measurements. If I am correct, this is a weakness of this paper.

*This is correct, however, the favored loss process, turbulent inertial deposition, is not very sensitive to aerosol size.*

Page 7 and page 8. The discussion of four possible loss processes. I think it is probably true that losses due to the physical behavior of the accumulation mode size aerosol are important and worth discussing in the spirit of a "first order estimate". However, beyond that, the situation is probably much more complex, particularly at different flow rates and with other possible, but less common for the sites studied, intake conditions such as "clean air" and a high intake fraction of unattached progeny. Ideally, redo the modelling using a two mode model. Otherwise, maybe shorten this section making clear the problem is too difficult to warrant more than a first order estimate in an experimentally oriented paper.

*As mentioned above, we de-emphasized the validity of our theoretical estimates and make clearer that our experimental results are only valid for this particular setting.*

Page 10. "The observed activity-dependency of the loss . . ." I suspect this faithfully measured and reported variation in the transmission efficiency is only the tip of the iceberg. The unexplained variation at low concentrations is probably due to something like clean air with lower aerosol concentrations and/or a higher unattached fraction of progeny at the tubing intake -- conditions more common at certain other meteorological towers. The present modelling and experimental protocol are inadequate to identify the exact cause, much less provide other researchers with the tools to adequately predict losses at a different site, with different meteorological conditions, with different flow rates, and with different tubing. This is why I recommend the following. If the present authors have the ability to make some the experimental and modelling revisions I suggest, that would strengthen the paper. Otherwise, I think their conclusions should be more carefully stated along the lines of "… We have made careful measurements for one particular set of test conditions and analyzed our results with a first order model. Given the significant progeny losses we observed with long sampling tubing, and unexplained variations in the loss fraction, we conclude that the approach of using long tubing intakes to measure atmospheric radon and Po214 is not presently reliable enough to recommend for situations where reference-grade measurements of atmospheric Rn are required." In addition, with a paper of more limited scope, it might be possible to shorten the paper and/or move more material to the supplement.

*We follow the suggestion of this reviewer, however, we kept the theoretical loss estimates in the main manuscript, as they, although by far not exhaustive, seem to support our experimental findings.*